# A Flexible Input Mapping System for Next-Generation Virtual Reality Controllers

**Eun-Seok Lee [1] and Byeong-Seok Shin [2,\*]**

[1] Department of VR Game Application, Yuhan University, Bucheon 14780, Korea; elflee77@gmail.com
[2] Department of Computer Engineering, Inha University, Incheon 22212, Korea
\* Correspondence: bsshin@inha.ac.kr; Tel.: +82-860-7452

**Abstract:** This paper proposes an input mapping system that can transform various input signals from next-generation virtual reality devices to suit existing virtual reality content. Existing interactions of virtual reality content are developed based on input values for standardized commercial haptic controllers. This prevents the challenge of new ideas in content. However, controllers that are not compatible with existing virtual reality content have to take significant risks until commercialization. The proposed system allows content developers to map streams of new input devices to standard input events for use in existing content. This allows the reuse of code from existing content, even with new devices, effectively reducing development tasks. Further, it is possible to define a new input method from the perspective of content instead of the sensing results of the input device, allowing for content-specific standardization in content-oriented industries such as games and virtual reality.

**Keywords:** virtual reality; haptic controller; game engine; event-driven system; framework





## 1. Introduction

In the field of virtual reality (VR), various devices have been proposed to provide a more realistic sense. Recently, research has been conducted on various input and output devices, such haptic controllers [1,2], motion controllers [3], and head mounted display (HMD) devices [4]. These input devices are referred to as controllers in VR equipment. Controllers provide user-to-content interactions in virtual reality in a variety of ways. The technology that detects user behavior in real-world space and converts it into digital signals has provided a content development environment for interaction through user movement in VR content. These methods [5,6] allow users to use acceleration sensors or depth cameras to turn gestures into input signals and to use them to develop content. Input methods using gesture recognition are widely used in modern PC and VR devices as well as in mobile phones and portable game consoles, such as those displayed in Figure 1.

The development of camera applications and the convergence of virtual reality technology have led to the integration of realistic images and the virtual universe [7]. With the fusion of these VR [8] and augmented reality [9] technologies, a concept called mixed reality (MR) [10] was proposed.

These techniques allow the HMD, an output device that visually provides virtual reality to users, to fuse images from cameras or reconstructed images from input devices, such as gyro sensors and acceleration sensors, to mix reality and virtual worlds. These attempts are mainly used for guidance in areas such as industrial sites and exhibitions and include Microsoft's HoloLens [11,12].

Recent studies have been active on how to implement touch using force, vibration, motion, etc., using devices such as haptic controllers [1], among virtual reality-based input devices. Haptic interface is a system that delivers a touch to the user. Haptic devices generate touch through direct user contact and physical delivery to the user. It communicates the movement of the happing device to VR, consisting of computer graphics and physical characteristics. In VR, the user's contact with the object is delivered to the

haptic device with a controller that conveys the touch, allowing the user to feel the texture. The ultimate goal of a haptic interface is to make users feel the same physical characteristics as a modeled virtual environment or a real environment through a haptic device. Haptic interfaces have agents in the VR world that recognize the change in location of the haptic device so that the device matches the VR environment. An ideal haptic interface system controls the force generated by a virtual agent hitting a virtual object and the force a person feels through a haptic device. The implementation of an ideal haptic interface requires the study of accurate modeling methods and hardware control methods that can implement those models.

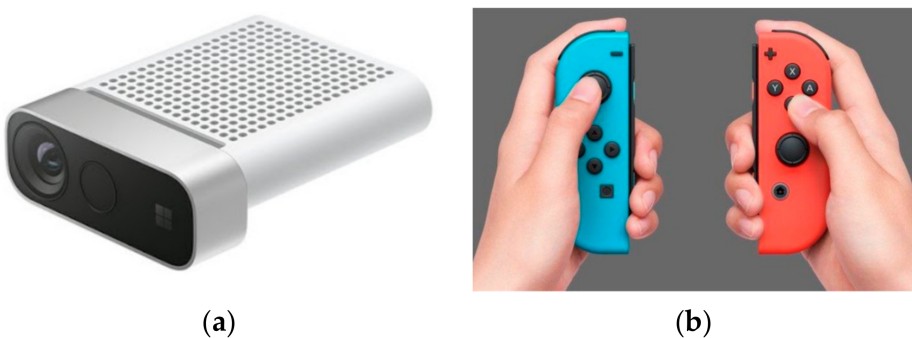

**Figure 1.** Devices that recognize user action. (**a**) Azure Kinect (**b**) Nintendo Joy-Con.

Haptic controllers are now used as representative input devices for VR devices. The development of these controllers provides an environment for developing more realistic VR content. Previous studies have developed methods of modeling touch and devices used in various forms, such as those presented in Figure 2, implementing vibrations using actuators [13], textured stimuli [14], and air injection and inhalation [15].

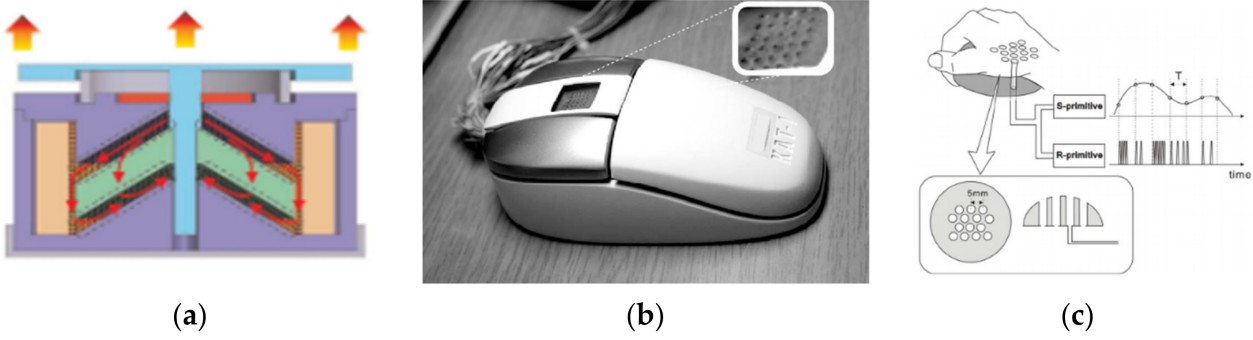

**Figure 2.** Various haptic devices. (**a**) Vibration using actuator (**b**) Texture stimulating mouse (**c**) Air Injection and Inhalation Devices.

If a standardized SDK for VR device is not provided, content developers cannot use the tools provided by the game engine used to create VR or game content. Therefore, in the case of a controller equipped with a new sensor, the manufacturer must directly implement and provide a SDK that works for each game engine. VR and games include not only engineering and scientific aspects, but also artistic aspects. In particular, the direction of content development is often determined by illogical factors such as artistic elements or public experience. These illogical elements are mainly standardized by the game engine and provided in the form of development tools or scripts.

However, many input devices do not support standardized software development kits (SDK), making it difficult to provide a development environment for content developers. This is because each device has a different protocol type of input signal.

When developing an SDK from the standpoint of device development, the SDK mainly serves to provide input signals generated from the device as a stream. However, in general,

game content is provided by applying various algorithms such as physics, lighting, and mathematics to the signal of the controller through the game engine [16,17]. It should be made to feel. Since it is not easy to standardize the method of expressing this subjective feeling, content developers are implementing it directly for each device [18]. Whenever various sensors in a game platform are applied to a game, a controller compatibility request is occurring. The proposed method proposes an input mapping system that will help facilitate this compatibility so that content developers can directly define input signals of newly developed input devices as well as general controllers provided by existing game engines. This can be virtualized so that existing content developers or middleware developers can use it without modifying the code according to the event of the tool provided by the module or engine created by the engine.

Defined input events themselves can be processed in real time in the proposed system. These events can be used in a form suitable for content development, unlike conventional device-friendly (prioritize performance) SDKs. Further, if the device used is changed to reimplement and map only to defined events in the system, even if the underlying SDK is changed, the new device can be easily used in existing content. For these mapping systems, middleware can easily overcome the challenges of existing content developers, and it clearly separates devices from content areas, making it easier to reuse code and effectively reducing the cost of rebuilding content for new devices.

Section 2 presents the input mapping system proposed in this paper. Section 3 presents the results of the experiment. We provide our conclusions in Section 4.

## 2. Input Mapping System

This section describes an input mapping system that makes it easier for content developers who produce content from VR, MR, and extended reality (XR) [18,19] to apply newly developed devices to content. Typically, SDKs in hardware are designed primarily to receive signals most efficiently from low-performance devices. Standard input devices have well-defined signals and are designed for easy use by content developers in middleware, such as VR Engine. However, devices that are not standardized have their own definitions of input signals, so they are not supported by middleware or have to be controlled directly against low levels.

### 2.1. Standard Input Device

The standard keyboard, mouse, and gamepad, which are typically available on PCs, provide standard application programming interface (API) in the operating system (OS). If the OS provides a standard API, content can be developed in the same way as shown in Figure 3. Standard APIs have been created to reflect the requirements of content developers and device developers for a long time. Thus, even in middleware, such as a game engine [17] where VR content can be produced, it is basically designed to control input devices using modules such as Input Manager.

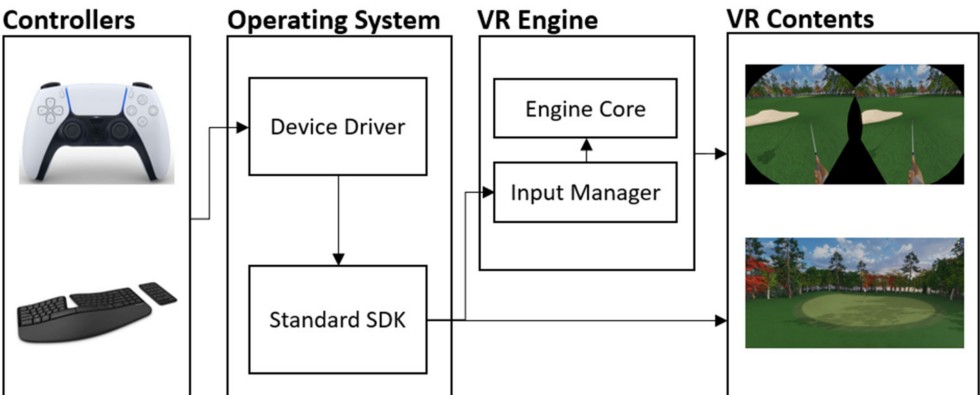

**Figure 3.** Content development flows when using standard input devices.

Thus, these standard input devices can be developed directly without significant difficulty using middleware, such as game engines, or implementing direct content. In the case of standard input devices, if the device is of the same type, there is no need to modify the source code in the content, even if the equipment is changed. However, if you deviate from the functionality of standard input devices, such as keyboards, mouse, and gamepads, you will not be able to fully utilize the functionality of the device.

## 2.2. Non-Standard Input Devices

Examples of non-standard devices include Leap motion [3], haptic controllers, light detection and ranging (LiDAR) [20], and VR/AR cameras [21,22]. These devices provide various signals to the user, such as images, three-dimensional coordinates, or 3D animations, as well as the input of buttons on the device. These devices communicate input signals to users through their own APIs, rather than standardized SDKs.

Figure 4 demonstrates how content is designed when non-standard devices are used. Game engines, commonly used as VR engines, provide dedicated plug-ins. These plug-ins allow the use of an input manager, a built-in module of the game engine. This can be found on devices such as Oculus rift VR [23,24] and Leap motion. These plug-ins can easily communicate the information needed in the content and process the information in the form that will be used in the content.

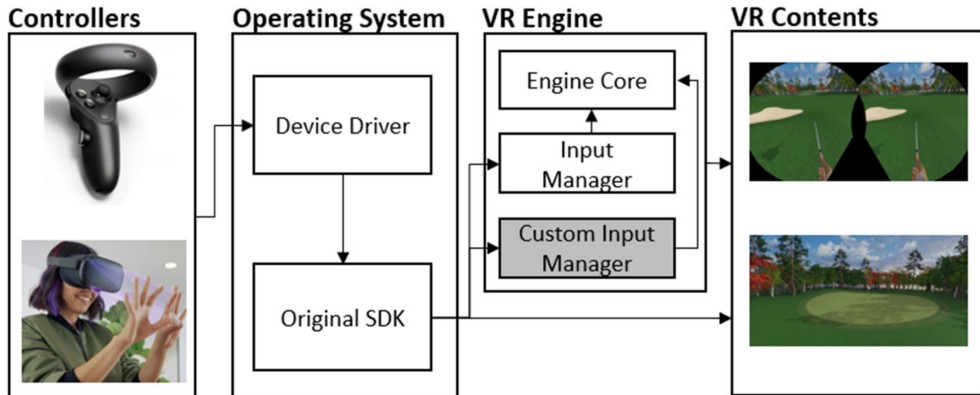

**Figure 4.** Content development flow when using non-standard input devices.

However, not all devices provide plug-ins for engines. If the plug-in is not provided, content developers must implement custom input manager directly on the engine using an SDK provided by the device, or use an SDK directly on the content without the engine. Generally, such non-standard SDKs are manufactured focusing on optimizing performance according to the propensity of equipment developers. In fact, such SDKs are not accessible because content developers have to implement the necessary functions directly from raw data if they are created without considering what they use in the content. This requires content developers to have knowledge of middleware, such as game engines, and the handling of devices.

## 2.3. Input Mapping System

In general, controller developers design APIs for development tools by delivering the numerical values of sensors obtained directly from hardware devices without processing. Therefore, in most cases, sensor data or image data require continuous values to be sent to the stream, and buttons and triggers are provided in the form of input events. However, since content manages code on a per-object basis, it is common to define interactions between objects in a particular event format.

The proposed method is a system that maps any controller development tool (SDK) to a user-defined form of signal to meet the requirements of content developers. In the proposed method, to reflect SDK features provided by the device in the content, we designed the system to act as event-driven, allowing content developers to invoke desired

actions in the form of events. The system uses a process called an input mapping daemon to handle input from the device. Figure 5 shows the input mapping daemon.

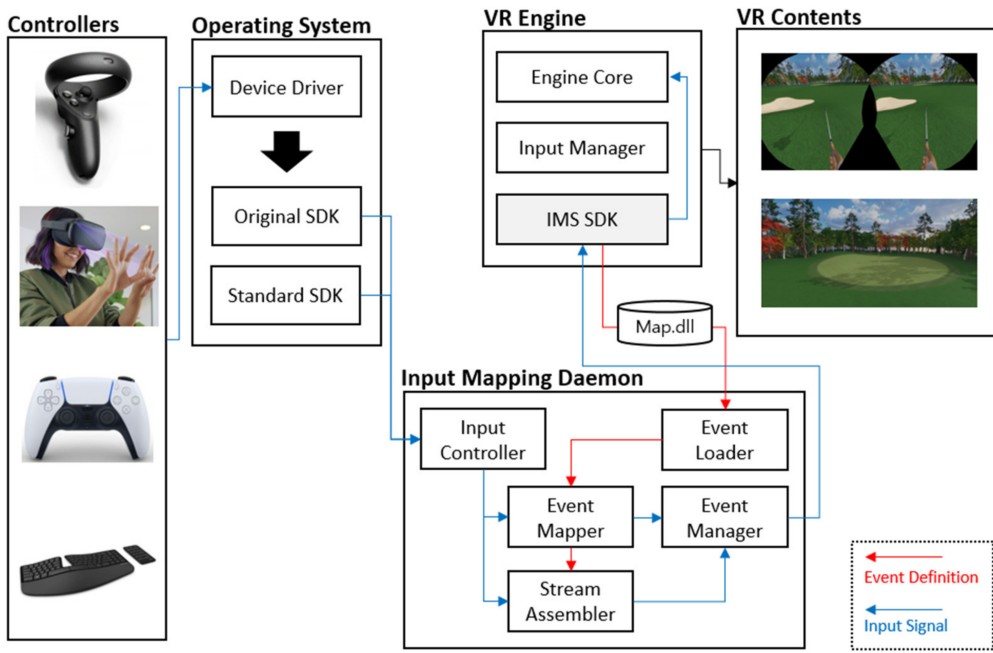

**Figure 5.** Overall architecture of the input mapping system.

Existing plug-in development as shown in Figure 4 should be implemented according to the set interface of the game engine. This has limitations in accepting new types of sensors or novel sensing ideas that are different from existing ones. Therefore, if you need to use these devices, you need to implement a new input module that can replace the input manager [25,26] of the VR engine. This has no choice but to develop as a subordinate to the VR engine. In this way, when an input module is manufactured and used externally, the priority of the engine core that executes the central command is lower than that of other major modules, which causes a problem of input lag in real-time applications.

Since IMS is composed of a separate process, it is not dependent on the VR engine, and events can be designed in the desired form and delivered directly to the VR engine. Therefore, input lag does not occur, and even if complex computation is performed, it is performed separately from the VR application, thereby reducing the impact on application performance.

Events can be directly defined before the daemon is executed. This event is divided into an event that the daemon receives from the engine and an event that the VR engine receives from the daemon. The handlers for this event are made to be processed asynchronously, and the handler is implemented by the user in advance in the dynamic linked library (DLL). This is an architecture that allows device developers or content developers to independently create SDKs and apply them directly to applications without the need to conform to the standards of the game engine. Therefore, as the user defines the event, various devices can be combined and used, which is a method not allowed by the existing VR engine. In addition, if the sensor value of a new type of sensor can be defined in the form of an input event, all can be converted into the input of the VR engine and used.

The proposed system was unified into one output for standard and non-standard SDKs to avoid reprogramming the input-related parts whenever the input device was replaced in content. Inside the VR engine, the input mapping system (IMS) SDK is an engine plug-in for the proposed system designed for easy installation and use by content developers. IMS is provided by registering and using event handlers to respond to user-defined events.

User-defined events are passed to the daemon in the process of initializing the IMS SDK. When the daemon first runs, it receives a (DLL) or a file that stores user-defined

events through the event loader. The file consists of events that will be used by users and event handlers that will be mapped to them. This information is passed to the event mapper to map the signal from the device to the event and to the stream-assembler to analyze the stream in logical units to initialize the module.

After initialization through the event loader, the daemon connects to the device through a module called the input controller. The connected equipment transfers data to the daemon in the form of data streams or input events. The daemon analyzes the data values of the device, maps them to user-defined events, and then forwards them to the user. This approach is not affected by the engine if a device-related error occurs, as it occurs in a separate process. Consequently, the content development domain can be free from errors in the device or errors caused by processing input data. This is easy to optimize and can effectively reduce the cost of content creation because it requires only the step of mapping signals from input devices into events without modifying the source code of the content domain. These tasks clearly separate the controller development area from the content development area while maintaining the expressive power of the content, and provide an independent development environment.

### 3. Experimental Results and Discussion

In this section, a single VR content was implemented to demonstrate the efficiency of the input mapping system. Experiments were conducted to compare performance and efficiency with that of not using an input mapping system by applying various equipment. The experimental environment was a personal computer equipped with the Intel 7th generation i5-7500 CPU, 16 GB DDR4 of main memory and the MSI GTX1650 video card with 4GB GDDR6 of graphic memory. The HMD device that conducted the experiment was the HTC Vive pro, Oculus rift [27]. The controller used a Vive controller [22], Vive tracker [28], Oculus controller, Microsoft Xbox One Controller (the standard gamepad), and a morphable haptic controller [29] that provided a non-standard SDK. Haptic controllers were adjustable in length and thickness, and weight-centered movement was possible, making them suitable for content that utilizes various equipment. The development content used in the experiment was representative content that utilized a haptic controller, and it was developed by selecting a golf simulator that required a variety of weights and grips. Golf content utilizes various clubs and can be played while adjusting the center of gravity, length, and thickness of each club. Figure 6 shows screenshots of the proposed golf simulation content.

Here, we conducted two comparative experiments to demonstrate the efficiency of the proposed system. The comparison targets were golf content developed using the input mapping system and golf content implemented directly through the SDK of middleware (Unity 3D) and devices (SteamVR). The first experiment was a performance comparison. Most of the source codes of the content were encapsulated to avoid duplicate development as much as possible, and device-related codes were written as much as possible so that there was no difference in performance depending on the developer's capabilities.

Table 1 defines the daemon's events to be received on the client side. The input map-ping system stores the event to occur depending on the device, the data to be received when the event occurs, and the ID of the event handler in the JSON-style file. Table 1 is a list of events defined in the JSON-style file. In this content, a button that allows the user to move to the point pointed to by the user is required, a button to call up or close a menu, and a haptic dedicated interface that changes the length, weight, and thickness of the hap-tic device according to changing the golf club are required. Therefore, as shown in Table 1, events such as trigger, menu button, change in length, thickness, and weight of the haptic device, state change such as power or connection, and position initialization (calibration) are defined.

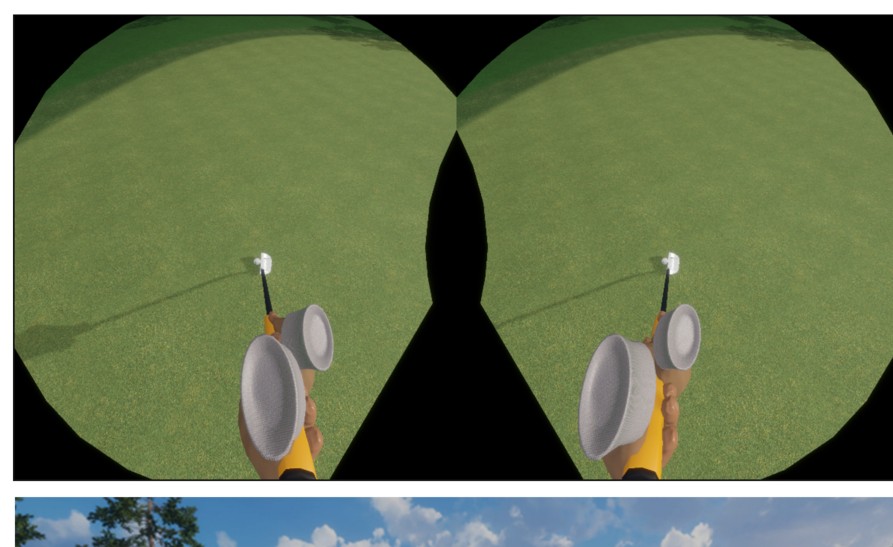

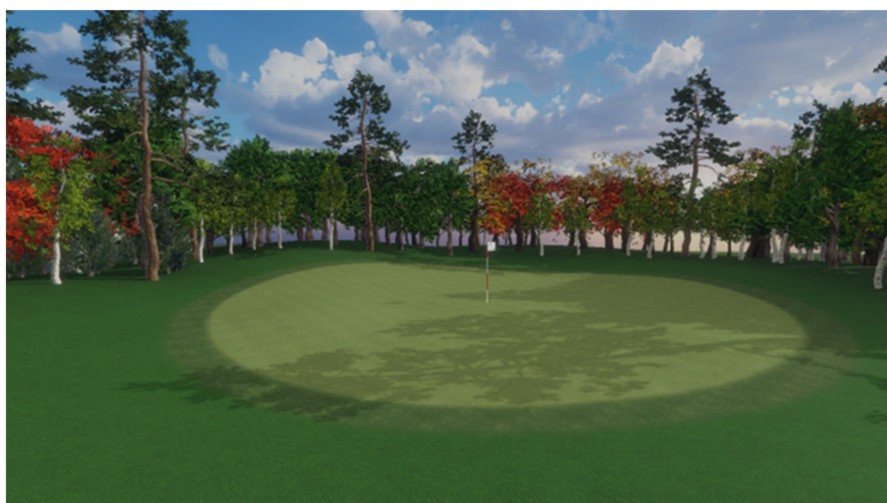

**Figure 6.** VR content for experiments.

**Table 1.** Client-side event list of contents.

| Event Name | Description | Contained Data |
|---|---|---|
| OnPosChange | Position of the controller changes. | float4 (vec4) |
| OnRotateChange | Axis and angle of controller's rotation. | float4 (vec4) |
| OnButtonPressed | Button Pressed. | none |
| OnButtonReleased | Button Released. | none |
| OnTriggerPressed | Trigger Pressed. | float |
| OnChangeLengthStart | Start changing the length of the controller. | none |
| OnChangeLengthEnd | The length change of the controller has ended. | none |
| OnChangeThicknessStart | Start changing thickness of controller. | none |
| OnChangeThicknessEnd | The thickness change of the controller has ended. | none |
| OnChangeWeightStart | A shift in the center of gravity of the controller has begun. | none |
| OnChangeWeightEnd | The change in the center of gravity of the controller has ended. | none |
| OnSendProperty | Returns the status value of the controller | string |
| OnInitialized | Initialized completed | none |

Since the daemon's side controls the device, it is necessary to perform not only signals received from the device but also commands requested by the client. Table 2 is the command used to receive the client-side's command on the daemon-side. The client can give the daemon a task in the form of a command. Using this, the client registers a command that

can control the device through the event loader, and receives a return in the form of an event after the command is performed.

**Table 2.** Daemon-side command list of contents.

| Command Name | Description | Data Type |
|---|---|---|
| Initialize | Initialize states. | none |
| GetProperty | Request property value. | string |
| ChangeLength | Request to change the length of the device. | int |
| ChangeThickness | Request thickness change of device. | int |
| ChangeWeight | Request weight change for device. | int |

Table 3 shows the results of the running time of the command on the client-side. The input mapping system has a relatively uniform command execution rate compared to the VR engine. This is because the Unity 3D engine used is single thread and implemented as coroutine, so it waits for the device. The initialization time is very long, especially for haptic devices. The proposed system is more efficient than engine-only methods because the daemon handles these tasks instead. The controller of Oculus rift and Vive pro were standardized through SteamVR, operated with the same code, and there was no difference in performance. Vive tracker also showed same performance. Therefore, in Table 3, these three are referred to as VR controllers and their performance is measured. The performance measurement scenario was as follows. After measuring the time to initialize the device and system from the contents used in the experiment, and performing the method of calling the device property values at once, 10 times, the processing speed was averaged. In the case of the haptic device, in the process of converting the approach club to the putter, the event processing time was measured 10 times and averaged.

**Table 3.** Comparison of the speed of command execution with the proposed method with the VR engine only (ms).

| Device Input | Proposed System | | | Unity 3D Only | | |
|---|---|---|---|---|---|---|
| | VR Controller | Gamepad | Haptic | VR Controller | Gamepad | Haptic |
| Initialize | 2 | 2 | 2 | 3 | 1 | 16 |
| GetProperty | 2 | 1 | 2 | 3 | 3 | 4 |
| ChangeThickness | × | × | 2 | × | × | 5 |
| ChangeWeight | × | × | 1 | × | × | 3 |
| ChangeLength | × | × | 1 | × | × | 4 |

Events provided by the game engine that are standardized such as position, rotation, button, and trigger acted immediately. Therefore, the operation time was similar, so no comparison was made. However, in the case of an event added by the user in the 'Initialization', 'GetProperty', 'ChangeLength', 'ChangeThickness' and 'ChangeWeight', there was a performance difference between the method and the proposed method.

The proposed system uses a multi-process model, which is not a fair comparison with the engine-only case because it is performed in a single thread environment. However, event-driven [24,25] approaches to responses from all commands and devices are more appropriate than non-VR content, which requires a screen to be rendered at a uniform rate. The proposed method handles all events asynchronously, so the sound of the screen rendering does not break due to the device.

The second experiment measured the number of times a client application crashed and terminated during a two-week bug fix. The second experiment measured the number of times the client application crashed and exited over two weeks of bug fixing. In this experiment, 57 specific behaviors to be tested for every 7 stages of the content were defined in the QA sheet in order to consider all special situations. A total of 285 actions and

37 cases for title and menu screens were combined, and through 322 test cases, if a crash occurred or an action different from the expected action was performed, it was considered a bug. These tests were measured once a week, and the number of bugs can be seen in Figure 7. The tests of both versions were conducted by the same person and same personal computer. Furthermore, the development and bug fixes were performed by 3 people in the same period.

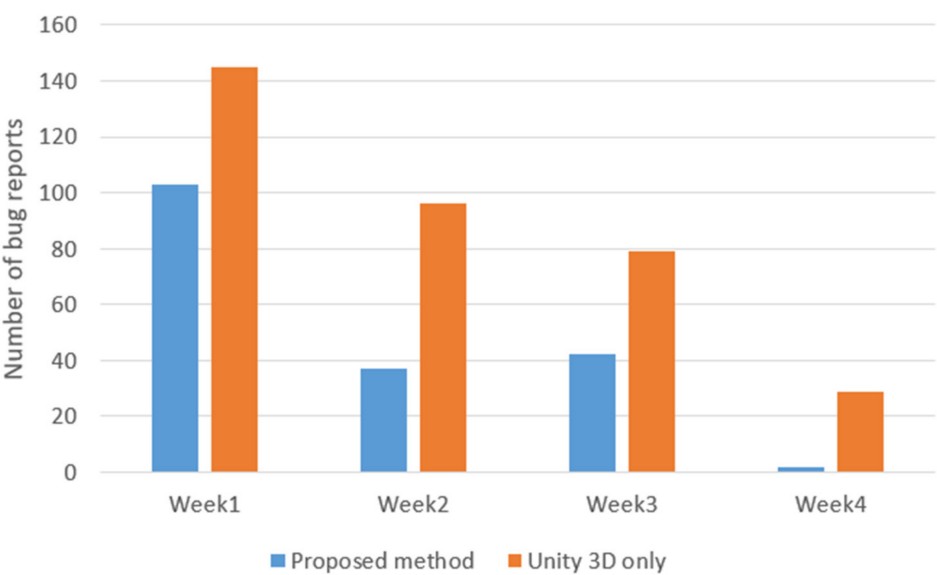

**Figure 7.** Comparison of the number of bug reports between Unity 3D and the proposed method.

As the results of the second experiment showed, the input mapping system clearly distinguished between client-side and device-side (daemon-side) development. Unlike the client side, the device side is concerned with optimization by processing tasks asynchronously, whereas the content domain is concerned with managing algorithms that manipulate objects and various objects, such as overall physics, lighting, and sound. If this different kind of code is combined together in one process, it can only cause difficulties when an unknown error occurs. Most of the errors occurred in the part that is asynchronously processed to process the device signal in the script of the game engine, because it is out of sync with the events generated by algorithms or objects, or the memory is not invaded or released. These mistakes are very difficult to debug because they are not accompanied by an error message. In the proposed method, it is much easier to troubleshoot errors because processes are clearly separated and the processing sequence of events can be monitored independently of the game engine. This helps to shorten the development speed and create higher quality applications as shown in Figure 7.

Therefore, the proposed method can avoid this confusion. Furthermore, the development period can be effectively reduced because only the device side can be isolated and focused on development when the device is changed.

## 4. Conclusions

In this paper, we propose an input mapping system that can efficiently apply various next-generation haptic controllers to VR content. It is a system that can define events from the perspective of content and create handlers that map to those events so that equipment that is not standardized can be easily applied to existing content.

The proposed input mapping system was able to effectively reduce errors that could occur when supporting a new non-standard type of input controller for already produced content through middleware. When defining standard content-only events through this input mapping system, content developers can perform input-related tests using a keyboard or virtual controller without a device. In addition, device developers can proceed with device debugging based on the log of events occurring in the content. This effectively

separates the roles of developers working in different environments, which can also help attach new types of controllers to content if only events are implemented. Therefore, the proposed system allows device experts to participate in numerical analysis and fine-tuning tasks that have been left up to existing content developers, making it easier for more equipment to stick to content without understanding the content.

For future work, it is necessary to build a system suitable for more diverse cases such as olfactory modality and wearable controllers so that content developers can more easily map input signals to controllers in different environments and perspectives.

**Author Contributions:** Conceptualization, E.-S.L. and B.-S.S.; methodology, E.-S.L.; software, E.-S.L.; validation, E.-S.L. and B.-S.S.; formal analysis, B.-S.S.; investigation, E.-S.L.; resources, E.-S.L.; data curation, B.-S.S.; writing—original draft preparation, E.-S.L.; writing—review and editing, E.-S.L.; visualization, E.-S.L.; supervision, B.-S.S.; project administration, B.-S.S.; funding acquisition, B.-S.S. Both authors have read and agreed to the published version of the manuscript.

**Funding:** This work was supported by INHA UNIVERSITY Research Grant.

**Acknowledgments:** This work was supported by INHA UNIVERSITY Research Grant.

**Conflicts of Interest:** The authors declare no conflict of interest.

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
