# Peer review of "A Flexible Input Mapping System for Next-Generation Virtual Reality Controllers"

_electronics, doi:10.3390/electronics10172149_

Round 1
Reviewer 1 Report
In the paper, the authors examine an input mapping system that can transform various input signals from next-generation virtual reality devices to suit existing virtual reality contents. The work is clear and accurate while being interesting, relevant to its research field. It may be suggested for publication after revision into journal format.
Author Response
Thank you for taking your valuable time to review this paper. The paper was corrected by native speakers through a professional calibration company, after reflecting the all reviewer's comments.
Reviewer 2 Report
This paper proposes an input mapping system from "next-generation" VR devices to standard inputs.
The paper is very hard to understand due to the low quality of the English writing. I assume the authors are not native English speakers and so it is understandable to have a lower quality English writing, but in this case this may have affected my evaluation of the paper. I would suggest that authors try to engage a native speaker to review the writing.
After reading the paper, I was left with several questions that I believe should be addressed.
The first question is the argument for the need of this input mapping system. To me, the argument is not very compelling. Authors state that "...controllers that are not compatible with existing virtual reality content have to take significant risks until commercialization." On the one hand, new devices appear every day and many make their way into commercial/consumer applications. So what is the problem, exactly? On the other hand, authors state that "map streams of new input devices to standard input events for use in existing content". We don't really want to use these new devices to interact in the same way we interact while using other standard devices. What is the point of using a Leap Motion device in replacement of a mouse? Of course there is a small gain (for the manufacturer) in emulating standard devices, but that is not the purpose of introducing a new device.
Authors state that "Input methods using gesture recognitions are widely used in modern PC and VR devices as well as mobile phones and portable game consoles, such as Figure 1." Is not this against your own argument? If these non-standard controllers are being widely used, what is the problem this paper is trying to solve?
"However, not all devices provide plug-ins for engines. If the plug-in is not provided, content developers must implement custom input manager directly on the engine using SDK provided by the device" Is not this a problem for the manufacturer to solve, if they wish to be successfull?
Another question is that authors do not review previous attempts at input mapping. There have been many proposals for input mapping in very different areas. These are completely ignored in this paper. I think authors should do a proper review of the literature and place their proposal within the vast literature that has proposed similar solutions. I see no novelty here. Not scientifically, nor from an engineering point of view.
Another question is about the design of the solution. There is no information about the reasoning for the proposed architecture. It is also not clear whether the list of events supported is fixed, or how one would go about to adding support for a new device.
A major issue is that the experiments do not validate the proposed system on the initial arguments. The experiments do not for example, allow controlling an existing VR application with a non-standard input device, without making changes to the existing application. It also does not show the difficulty of adding support for a completely new device. It seems the proposed system is simply shifting the programming effort from developing an addon for a specific device, to reprogramming the middleware.
Other, less important issues (in light of the previous points):
- Many sentences are hard/impossible to understand. Just a few examples:
- "Existing virtual reality content is developed based on factor values for standardized commercial haptic controllers" What are factor values or factor-driven API? This expression is used in the abstract, and then only in the conclusions. It seems an important aspect that is not mentioned in the introduction or when describing the proposed system.
- "The technology that detects user behavior in real-world space and converts it into digital signals has provided a content development environment for interaction through user movement in virtual reality content." What is a content development environment?
- "Haptic interfaces are often used in the same sense as haptic devices because they are limited to hardware in a narrow sense" I did not understand the meaning of this sentence.
- "However, these various input devices are standardized SDKs" Did not understand the meaning.
- "the necessary APIs for devices in a content-oriented manner." What is content-oriented manner?
- In experiments, it is not clear how the performance was measured.
- Table 1 is not described. It is not clear what the API of the device is, and how it is mapped into standard events.
In summary, the argument is not compelling, there is no review of related work, the proposed system is not well described, and the experiments do not seem relevant for the problem that was being addressed.
Author Response
Thank you for reviewing the paper. I have uploaded a reply to the review as an attachment.

Reviewer 3 Report
The manuscript is focused on the development of a system to help during the integration of VR controllers into engines to display content. The subject is interesting and appropriate. The text is well written and accessible, however, along with the manuscript (attached .pdf file), some comments were made to clarify doubts and some suggestions were placed.

Author Response

(The authors gave the same response as above.)

Reviewer 4 Report
The article is timely and interesting. After reading it, the reviewer's first though were: was this only tested on the Vive? The reviewers believes that more HMDs should be included for a better investigation. At the moment, perhaps the authors should rename the article, because its current title suggests that this method works with all HMDs. Naturally, it is a possibility, but it was not tested.
Overall, the English is good, readable, however, moderate changes required.
The methodology of the study should be explained in more detail. For example, present the developed golf game and the mapping procedure in more detail: i.e., what are the controls on the Vive? How are these controls mapped to the tested devices (meaning, which button is which)? Can they be customized?
Regarding English and methodology, the reviewer chose this sentence for multiple remarks: "personal computer equipped with the Intel 7th generation i5 and 16GB of main memory and the GTX1650 video card" <- the word "CPU" is missing after i5. What is the speed of the CPU and memory? Is the memory type DDR3, DDR4, etc.? How much is the virtual memory of the GTX1650 card? Which GTX1650 card? There are multiple versions of it.
The format of references is good.
Some smaller remarks:
HMD is short for "head-mounted display", not head mount display
Figure 1 (a) is named Azure Kinect
Although it is natural for scientists in this field, but the abbreviations of VR, MR, XR LiDAR are not defined in this article. They should be included.
Please define what is a "morphable haptic controller that provide a nonstandard SDK" exactly (lines 181 - 182).
Tables should be placed after they are mentioned in the text.
Author Response

(The authors gave the same response as above.)

Round 2
Reviewer 2 Report
The authors did a substantial effort in revising the paper. Although I still think that a proper Related Work section would improve the contribution of the paper and a better description of the system could be provided, I believe it is now in a form that can be published.